# Abrupt warming and alpine glacial retreat through the last deglaciation in Alaska interrupted by modest Northern Hemisphere cooling

Joseph P. Tulenko[1,2], Jason P. Briner[2], Nicolas E. Young[3], and Joerg M. Schaefer[3]

[1]Berkeley Geochronology Center, 2455 Ridge Road, Berkeley, CA, USA 94709
[2]Department of Geology, University at Buffalo, 126 Cooke Hall, Buffalo, NY, USA 14260
[3]Lamont-Doherty Earth Observatory, Columbia University, 61 Route 9W, Palisades, NY, USA 10964

*Correspondence to*: Joseph P. Tulenko (jtulenko@bgc.org)

**Abstract.** Alpine glacier-based temperature reconstructions spanning the last deglaciation provide critical constraints on local-to-regional climate change and have been reported from several formerly glaciated regions around the world yet remain sparse from high northern latitude regions. Using newly and previously [10]Be-dated moraines, we report paleo-glacier equilibrium line altitudes (ELA) for 15 time slices spanning the Last Glacial Maximum (LGM) to the Little Ice Age (LIA) for a valley in the western Alaska Range. We translate our ELA reconstructions into a proxy for summer temperature by applying a dry adiabatic lapse rate at each reconstructed ELA relative to the outermost LIA moraine. We observe ~4°C warming through the last deglaciation at our site that took place in two steps following initial gradual warming: ~1.5°C abrupt warming at 16 ka, ~2 kyr after the onset of global $CO_2$ rise, and ~2° C warming at ~15 ka, near the start of the Bølling. Moraine deposition and modest summer cooling during Heinrich Stadial 1 and the early Younger Dryas (YD) suggest that despite these events expressing more strongly in wintertime, the classic blueprint of North Atlantic climate variability extends to the western Arctic region.

## 1 Introduction

Global mountain glacier recession is one of the clearest indicators of modern planetary warming (Hugonnet et al., 2021). Paleo-data suggest that alpine glaciers are highly sensitive to temperature changes on both centennial (Roe et al., 2017) and millennial time scales (Schaefer et al., 2006). The last deglaciation (~19 – 11 ka) contains periods of abrupt warming akin to contemporary climate change – particularly later in the interval during the 'late glacial' ~15-11 ka – and considerable effort has been spent to constrain the timing and pace of mountain glacier fluctuations world-wide through this interval (e.g., Putnam et al., 2013; Ivy-Ochs, 2015; Palascios et al., 2020). However, mountain glacier fluctuations in the high northern latitudes – in locations where ice sheets were largely absent or restricted– are less well characterized due in part to the limited availability of suitable sites for organic radiocarbon dating, which presents a significant challenge compared to more temperate regions (Wittmeier et al. 2020).

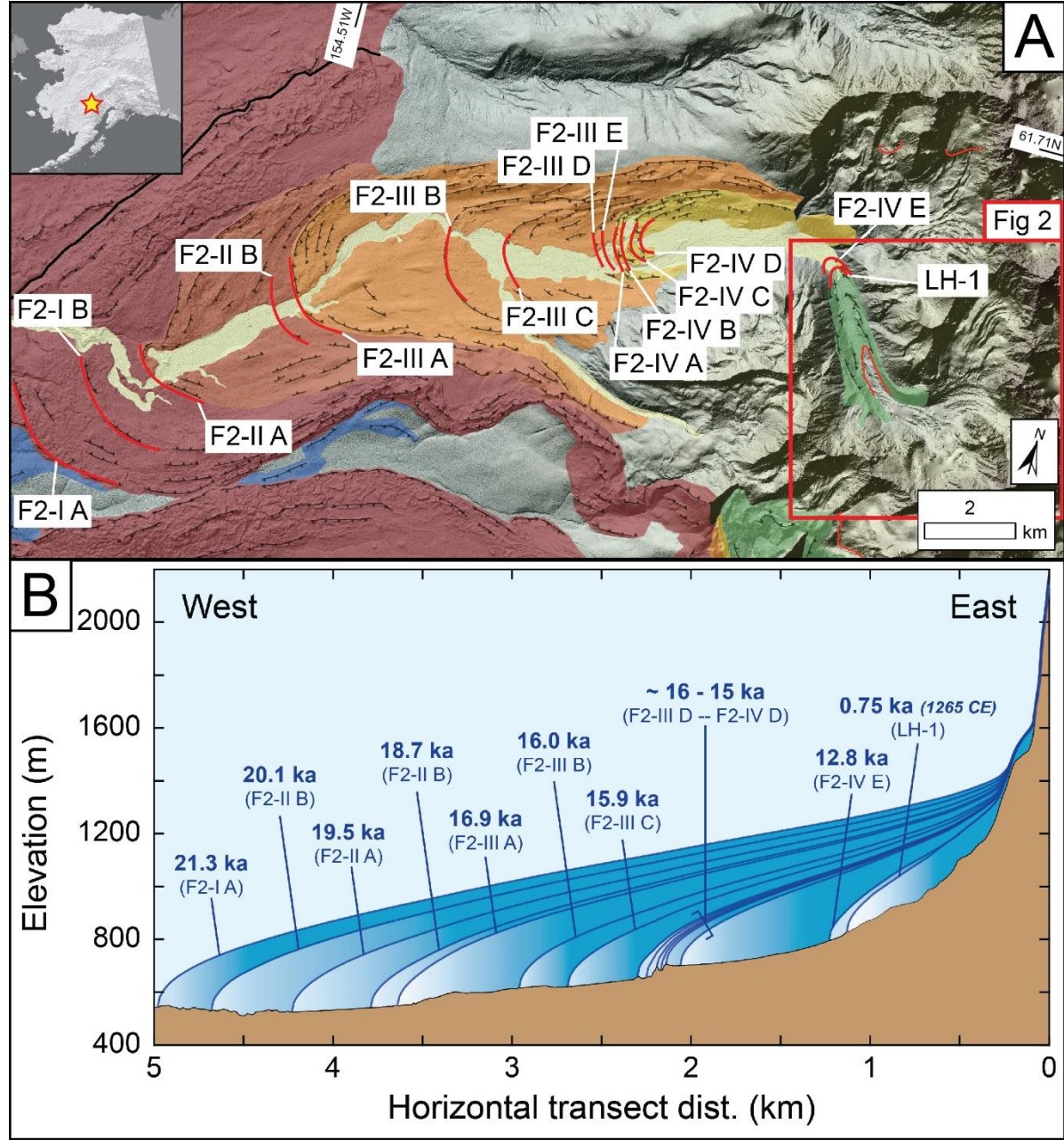

30

**Figure 1. A: Moraine map of the North Swift River valley with reconstructed moraine positions denoted with red lines (n = 15). Colour overlays denote broad morphostratigraphic equivalence of landforms and are arranged as darker shades being stratigraphically older than lighter shades. Blue shades are MIS 4-age moraines, brown/red shades into yellow shades are LGM and deglacial moraines, and green are Holocene-age moraines. Inset: starred location of the Revelation Mountains in Alaska. B:**
35 **Schematic of alpine glacier retreat in the North Swift River valley that includes moraine ages reported here and in Tulenko et al. (2022). Panel A source: ArcticDEM product available from the University of Minnesota Polar Geospatial Center. Inset map source: IBCAO product available from GEBCO.**

Glacier equilibrium line altitudes (ELAs) are a representative measure of the climatic conditions supporting a glacier. Climate-driven positive and negative mass balance changes force the ELA to lower or rise, respectively (Ohmura et al., 1992; Oerlemans et al., 2005, Braithewaite, 2008). Reconstructing paleo glaciers through detailed mapping and dating techniques provides valuable data for estimating paleo-ELAs, which serve as a powerful paleoclimate proxy for glaciated regions. Improved access to accurate digital elevation models (DEMs) and the development of semi-automated paleo-glacier surface and ELA reconstruction tools (Pellitero et al., 2015, 2016) paves a path for more objective measures of paleo-ELA calculation. Changes in ELA are dependent primarily on summer (JJA) temperatures and annual snowfall (Oerlemans et al., 2005). Some studies suggest, however, that in most environments other than the most extremely arid environments such as modern-day inner Mongolia, alpine glacier ELAs are dominated by summer temperatures (e.g., Rupper and Roe, 2008).

Building on a recently published [10]Be chronology of 14 moraines in the North Swift River valley, Revelation Mountains, western Alaska Range (Tulenko et al., 2018, 2022), we constrain additional [10]Be ages for three late Holocene moraines in the valley and generate a paleo-ELA based proxy summer temperature record from our site. We estimate the timing and pace of warming through the last deglaciation by transforming ΔELAs into summer temperature assuming a dry adiabatic lapse rate of 10°C/km. The magnitude of temperature change we obtain is supported by independent climate proxy records in Alaska, and we compare our record with other glacier-based temperature reconstructions to speculate about the controls on climate in Alaska through deglaciation.

## 2. The Revelation Mountains field site

The Revelation Mountains field site is located on the western limb of the Alaska Range (Fig. 1 inset) where dense sequences of moraines deposited by alpine glaciers fluctuating through the last deglaciation are well-preserved. The Revelation Mountains are cored by a granitic pluton, and large tabular boulders deposited on moraines are suitable targets for cosmogenic [10]Be exposure dating. The site is located far south from seismic activity along the Denali Fault, and thus moraines deposited in these valleys have likely remained more stable after deposition compared to moraines in locations more proximal to the Denali Fault system (e.g., Briner et al., 2005; Matmon et al., 2006; Dortch et al., 2010). All these factors have led to productive chronologic constraints on deglaciation in Alaska (Tulenko et al., 2022).

Within the Revelation Mountains, the North Swift River valley hosts the densest and most comprehensive sequence of moraines of any valley surveyed at our site (Fig. 1). For more details on the mapping and dating approaches used to generate the deglacial chronology in the North Swift River valley site, readers are referred to Tulenko et al. (2022). After a thorough moraine dating campaign of deglacial moraines in the valley, we additionally mapped and dated moraines in the valley deposited near the terminus of the extant glacier that were likely late Holocene in age (Fig. 2). For this study, the purpose was

70 to target late Holocene moraines to contextualize the magnitude of deglacial retreat in the North Swift River valley. Thus, all inferred climate fluctuations reported here are relative to the late Holocene advances observed at our site.

**3 Cosmogenic $^{10}$Be exposure dating of late Holocene glacier fluctuations in the Revelation Mountains**

To complete the time series of glacier retreat in the North Swift River valley, we collected surface samples in summer 2019 for $^{10}$Be dating from 15 granitic boulders on three distinct moraine crests between the innermost late glacial moraine (F2-IV

75 E) and an unnamed extant glacier (Figs. 1 and 2). The moraines are sharp-crested, and clast supported with minimal vegetation cover (Fig. 3). We selected the largest, tabular boulders (>1-2 m tall) with horizontal-to-sub-horizontal top surfaces to avoid excessive snow cover. Surface samples approximately 2 cm thick (see Table 1 for exact thickness measurements) and ~1 kg in mass were extracted using a Hilti brand angle grinder with diamond-tipped cutting discs and a hammer and chisel. GPS coordinates were taken with a handheld GPS, and shielding corrections were calculated using in-field measurements with a

80 handheld clinometer and the shielding correction calculator from the online exposure age calculator website (https://hess.ess.washington.edu/). We targeted large, tabular boulders on clast supported moraines to minimize the likelihood that post-depositional processes – e.g., snow cover and moraine degradation – inhibited production of cosmogenic $^{10}$Be in sampled surfaces, which would produce ages younger than the true timing of moraine deposition.

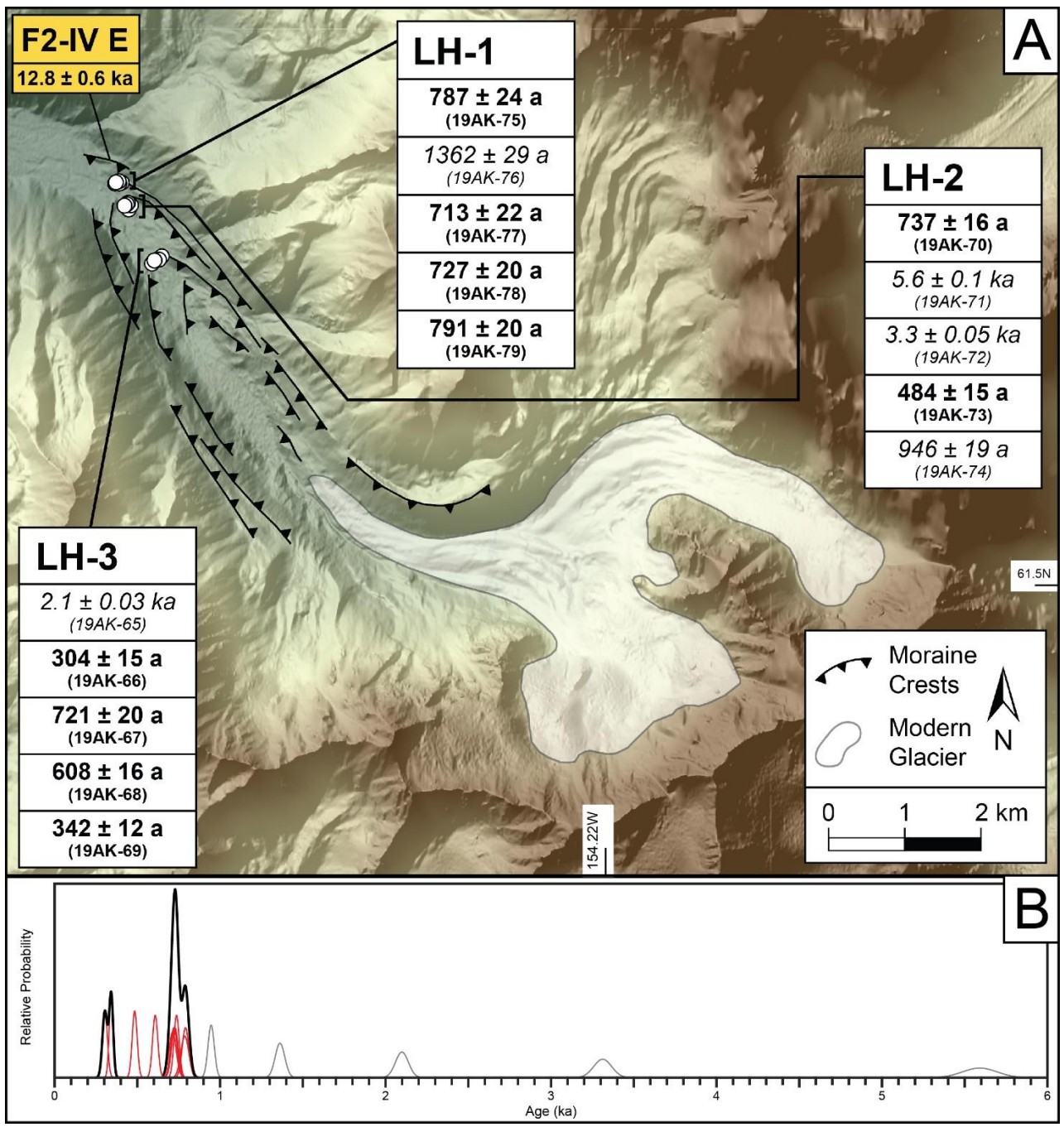

Figure 2. A) Shaded relief map of the modern North Swift River valley glacier and late Holocene moraines with [10]Be ages displayed. Samples listed in italics are suspected outliers. Age for the F2-IV E moraine segment from Tulenko et al., 2022. Map source: 5m resolution IFSAR Alaska DEM product available from the USGS. B) Normal probability density functions (pdfs) for individual ages in red and summed pdfs for the two clusters of ages around 750 a and 320 a in black. Ages older than expected and well-beyond two standard deviations from other ages in the dataset in grey.

All samples were processed from full surface sample to prepared [10]Be targets at the Lamont-Doherty Earth Observatory (LDEO) cosmogenic dating laboratory following procedures outlined in Schaefer et al. (2009). Beryllium measurements for samples and batch blanks were made at the Lawrence Livermore National Lab Center for Accelerator Mass Spectrometry (LLNL-CAMS). All [10]Be/[9]Be ratios were standardized to the 07KNSTD standard of 2.85 x 10[-12] (Nishiizumi et al., 2007), and blank-corrected final [10]Be concentrations are reported in Table 1 and in the ICE-D database (www.ice-d.org) along with all other relevant sample observations and measurements.

**Table 1. [10]Be age information**

| Sample name | Latitude (DD) | Longitude (DD) | Elevation (masl) | Thickness (cm) | Shielding correction | Qtz Dissolved (g) | Carrier Added (g) | [Be-10]/[Be-9] (10[-14]) | [Be-10] Atoms/g | Age (a) | Age (CE) |
|---|---|---|---|---|---|---|---|---|---|---|---|
| LH-1 | | | | | | | | | | | |
| 19AK-75[a] | 61.67542 | -154.28345 | 822 | 1.85 | 0.964221 | 53 | 0.1815 | 3.013 ± 0.092 | 7106 ± 217 | 787 ± 24 | 1232 ± 24 |
| 19AK-76[a] | 61.67548 | -154.28334 | 822 | 2.41 | 0.964221 | 48 | 0.182 | 4.661 ± 0.099 | 12194 ± 259 | 1362 ± 29 | 657 ± 29 |
| 19AK-77[a] | 61.67552 | -154.28322 | 818 | 4.92 | 0.964221 | 70 | 0.1822 | 3.509 ± 0.106 | 6259 ± 189 | 713 ± 22 | 1306 ± 22 |
| 19AK-78[a] | 61.67542 | -154.28313 | 820 | 2.52 | 0.964221 | 70 | 0.1819 | 3.637 ± 0.101 | 6521 ± 182 | 727 ± 20 | 1292 ± 20 |
| 19AK-79[a] | 61.67546 | -154.28278 | 820 | 2.87 | 0.964221 | 62 | 0.1821 | 3.526 ± 0.087 | 7071 ± 175 | 791 ± 20 | 1228 ± 20 |
| LH-2 | | | | | | | | | | | |
| 19AK-70[a] | 61.6741 | -154.28181 | 829 | 2.5 | 0.964221 | 70 | 0.1806 | 3.761 ± 0.081 | 6658 ± 143 | 737 ± 16 | 1282 ± 16 |
| 19AK-71[b] | 61.6739 | -154.28194 | 829 | 2.9 | 0.964221 | 71 | 0.1814 | 28.607 ± 0.532 | 50099 ± 931 | 5591 ± 104 | -3572 ± 104 |
| 19AK-72[b] | 61.67412 | -154.28212 | 828 | 2.5 | 0.964221 | 70 | 0.182 | 16.567 ± 0.269 | 29637 ± 481 | 3312 ± 54 | -1293 ± 54 |
| 19AK-73[a] | 61.6741 | -154.28217 | 828 | 2.2 | 0.964221 | 70 | 0.1822 | 2.433 ± 0.074 | 4375 ± 133 | 484 ± 15 | 1535 ± 15 |
| 19AK-74[a] | 61.67403 | -154.28236 | 832 | 2.67 | 0.964221 | 72 | 0.1807 | 4.940 ± 0.099 | 8563 ± 173 | 946 ± 19 | 1073 ± 19 |
| LH-3 | | | | | | | | | | | |
| 19AK-65[b] | 61.67059 | -154.27878 | 864 | 2.77 | 0.964245 | 71 | 0.1813 | 11.047 ± 0.206 | 19445 ± 362 | 2099 ± 39 | -80 ± 39 |
| 19AK-66[b] | 61.67073 | -154.27852 | 863 | 2.83 | 0.964245 | 69 | 0.1812 | 1.568 ± 0.077 | 2830 ± 140 | 304 ± 15 | 1715 ± 15 |
| 19AK-67[b] | 61.67073 | -154.27855 | 862 | 2.04 | 0.964245 | 69 | 0.1822 | 3.715 ± 0.104 | 6729 ± 188 | 721 ± 20 | 1298 ± 20 |
| 19AK-68[b] | 61.67081 | -154.27794 | 862 | 2.82 | 0.964245 | 71 | 0.1823 | 3.182 ± 0.082 | 5635 ± 145 | 608 ± 16 | 1411 ± 16 |
| 19AK-69[b] | 61.67096 | -154.27766 | 865 | 2.38 | 0.964245 | 73 | 0.1816 | 1.869 ± 0.065 | 3193 ± 112 | 342 ± 12 | 1677 ± 12 |

Table 1 Notes: [a]Blank ratio for these samples is 7.685E-16. [b]Blank ratio for these samples is 5.311E-16. Carrier concentration for all samples and blanks was 1029.6 ppm. Rock density assumed to be 2.65g/cm3 for all samples. Zero surface erosion

assumed. Ages presented here are calculated using the Arctic Production Rate (Young et al., 2013) and time-dependent scaling from Lal (1991)/Stone (2000).

[10]Be surface exposure ages reported here were calculated on the online exposure age calculator (Balco et al., 2008) using the Baffin Bay Arctic Production rate (Young et al., 2013), and the time-dependent scaling scheme from Lal (1991) and Stone (2000). Several previous studies in Alaska have utilized this production rate calibration/scaling scheme combination (Valentino et al., 2021; Tulenko et al., 2022 and references cited therein), and the production rate measured in Baffin Bay is statistically identical to other high-latitude Northern Hemisphere calibration sites such as the Scottish Highlands Rannoch Moor site

(Putnam et al., 2019), The Swiss Alps Chironico Landslide site (Claude et al., 2014), and Northeastern North America (Balco et al., 2009).Use of other commonly applied production rate calibrations would result in ages offset by ~3%. We do not attempt to correct for snow-cover. We also do not correct for boulder surface erosion since the boulders are crystalline granitic lithology and thus resistant to weathering, particularly on the brief (hundreds of years) period between deposition and sample measurement. Ages from this study converted to common era (CE; below) use 2019 (date of collection) as the reference starting

date.

Ages from the three dated moraines situated 200-1000 m inboard of the innermost late Pleistocene (F2-IV E) moraine range from $791 \pm 20 - 304 \pm 15$ a (1228 – 1715 CE (n = 10; not including 5 potential outliers; Fig. 1, Fig. 2, Table 1). Ages that are suspected outliers do not overlap with any other ages within two standard deviations and are older than the remaining ages

(Fig. 2b), which indicates they may have been influenced by nuclide inheritance. The outermost moraine (LH-1) has four ages that overlap within 2 standard deviations and average $755 \pm 35$ a ($1263 \pm 35$ CE). The next moraine (LH-2) does not have any ages that overlap, but one boulder age at $737 \pm 15$ a ($1282 \pm 15$ CE) coincides with the ages from LH-1 so it is possible it was deposited inboard of the LH-1 position at the same time the LH-1 moraine formed and then was recycled during the advance that formed LH-2. For the innermost moraine dated, two of the five boulder ages overlap within 2 standard deviations and

average $323 \pm 19$ a ($1696 \pm 19$ CE). Additionally, LH-3 contains one boulder age that also conforms well with the ages from LH-1 at $721 \pm 20$ a ($1298 \pm 20$ CE) and could have a similar history to the age from the LH-2 moraine at $737 \pm 15$ a ($1282 \pm 15$ CE).

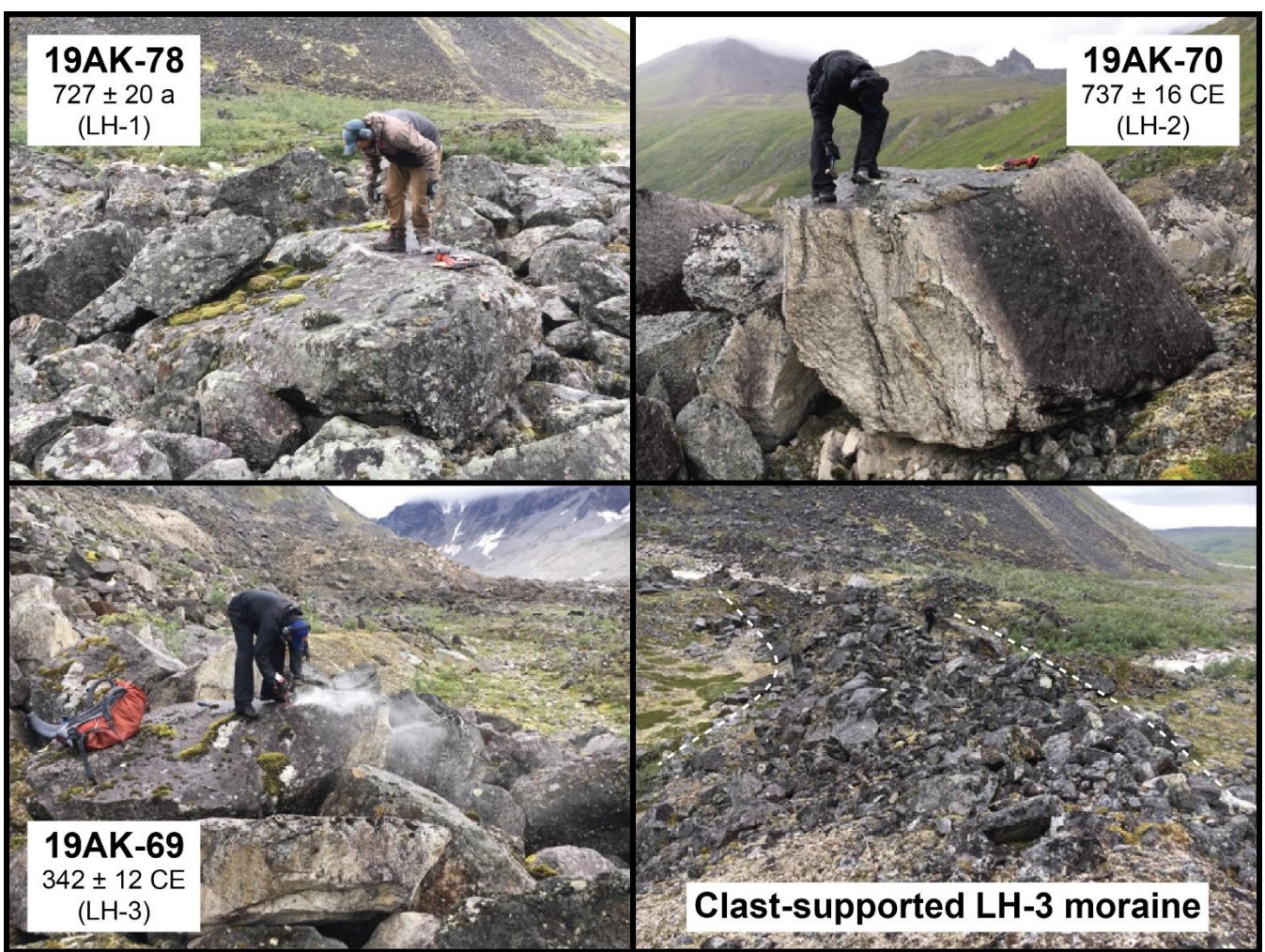

Figure 3. Example photos of sampled boulders on the three late Holocene moraines (top and bottom left panels). Bottom right panel is an image of the left-lateral segment of LH-3 where sampled boulders are situated. Note person on moraine crest for scale. Images of all sampled boulders can be found in the ICE-D database (www.ice-d.org).

Many of the well-constrained observations of Little Ice Age (LIA) glacier fluctuations in Alaska come from southern Alaska,
where tree-ring chronologies from trees sheared at their bases by glacial overriding tightly constrain episodes of glacier advance (Figs. 4 and 5; Wiles et al., 2002, 2004; Barclay et al., 2009). These precise chronologies reveal two major glacier advances at ~1200-1300 CE and ~1600-1800 CE (Fig. 5). In comparison, we observe moraines deposited in the North Swift River valley at $755 \pm 35$ a ($1263 \pm 35$ CE) and $323 \pm 19$ a ($1696 \pm 19$ CE). The coherence of $^{10}$Be ages in the North Swift River valley with tree-kill dates in southern Alaska and suggests similar climate controls for glaciers in the western Alaska Range
and in southern Alaska during the LIA.

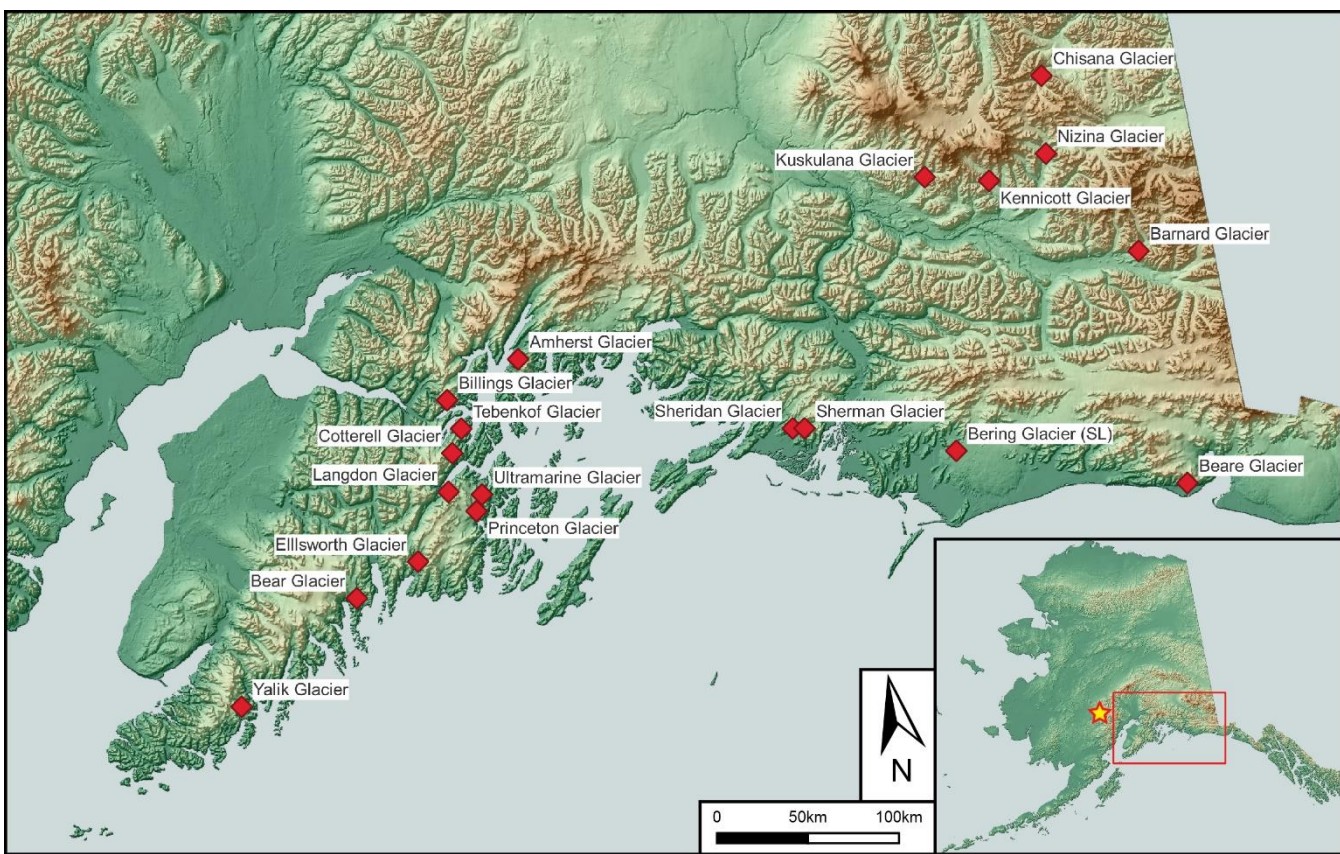

**Figure 4. Locations of glaciers in Southern Alaska with well-constrained late Holocene advances from tree-kill dates. Location of the Revelation Mountains denoted by a star in the inset. Basemap available from ASTER.**


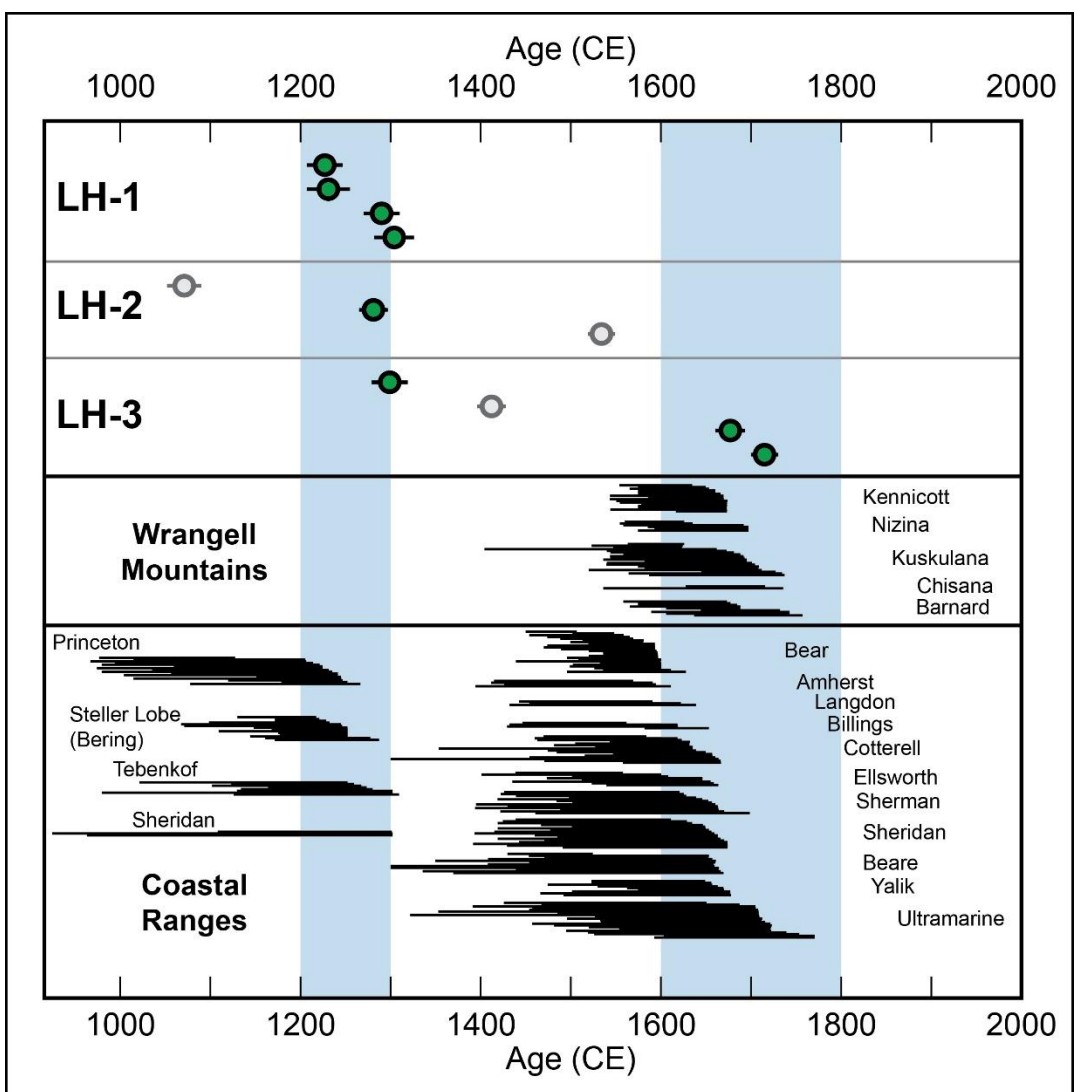

**Figure 5. Comparison of late Holocene moraine ages (green dots; grey dots are outliers) with tree-ring-based southern Alaska glacial fluctuations (black bars; Barclay et al., 2009). Light blue vertical bars indicate periods of moraine deposition in the Revelation Mountains (upper panel) and tree-kill dates signifying glacier culminations in Southern Alaska (lower panels) at ~1200-1300 CE and ~1600-1800 CE. See Fig. 4 for geographic locations of tree-kill dates.**

While the sampling techniques discussed above greatly reduce the likelihood of post-depositional processes impacting age determinations, it is virtually impossible to detect in the field which samples may have excessive cosmogenic nuclide inventories (i.e., nuclide inheritance) from surface exposure prior to glacier advances and moraine building events. Commonly with erosive (non-cold based) alpine glaciers, measured samples on moraines impacted by nuclide inheritance manifest as extreme values that do not conform with any other ages in the distribution (e.g., Heyman et al., 2011). Thus, we suspect all samples observed on our three moraines with extreme old values that do not conform with any other ages were impacted by

nuclide inheritance. Not considering the older extreme values across the three moraines (n = 5), we find 8 of the remaining 10 samples overlapping in two clusters, ~1200-1300 CE (n = 6) and ~1700 CE (n = 2). We tentatively conclude that because these samples are both internally consistent and consistent with other independent records of LIA glacier fluctuations in Alaska, they may reflect the true age of two LIA glacier advances at our site.

## 4 Glacier surfaces, ELAs and climate reconstructions in the Revelation Mountains through deglaciation

To construct a climate record built on the completed chronology at our site, we employ the tools developed for use in ArcGIS from Pelitero et al. (2015, 2016) to first reconstruct paleo-glacier surfaces and then equilibrium line altitudes (ELAs) for each of the previously dated moraine positions (n = 14) and the outermost late Holocene moraine position (LH-1; 15 total moraines). We select the outermost late Holocene moraine because it is the most robustly dated and the late Holocene moraines are all within 1 km of each other, resulting in similar paleo ELA values. The glacier surface reconstruction tool (GLARE; Pelitero et al., 2016) is semi-automated but requires the following inputs: a base map digital elevation model (DEM), polygons of maximum glacier outline constraints, and glacier center flowline polygons (Fig. 6). We mapped glacier outlines on a 5-meter resolution IFSAR DEM basemap provided by the USGS and available for download at the National Map (https://apps.nationalmap.gov/downloader/). Maximum constraints for each individual paleo-glacier position were outlined based on terminal and lateral moraines originally mapped in Tulenko et al., (2018), and extending up to the valley drainage divide in the steeper terrain once moraines were no longer visible. In steeper terrain, it is difficult to find preserved evidence of glacier extents, so the divide offers the maximum possible limit of glacial extent. For the remaining steps, we resampled the 5-meter resolution DEM down to 100-meter resolution to smooth out the surface. We suggest that since post-glacial landscape evolution has likely altered the land surface at fine scale, a smoothed surface may minimize any influence of post-glacial landscape evolution on our glacier surface reconstructions. Using the resampled DEM and the series of watershed analysis tools in ArcGIS, we produced ice flowline polylines that generally follow the modern drainages (Fig. 6). The GLARE tool then calculates ice thickness along the center flowline assuming perfect plasticity for ice rheology and a value for basal shear stress that we set to 100 kPa for all paleo-glaciers at all positions along their center flowlines. Finally, we use the topo to raster interpolation method (see Hutchinson et al., 2011 for details) supplied by the GLARE tool to reconstruct a glacier surface extending outward from center flowline thickness values of each paleo-glacier and confined either by their respective glacier outline where moraines exist or the valley wall in steeper terrain (see fig. 6 insets for examples).

Following glacier surface reconstructions, we employ the Accumulation Area Ratio method supplied by the ArcGIS ELA tool (Pelitero et al., 2015) for each paleo-glacier. We assume a ratio of 0.63 ± 0.04, which has been suggested as representative for alpine glaciers greater than 4 km$^2$ (Kern and Laszlo, 2010). ELA values for each glacier position, along with age constraints reported in Tulenko et al. (2022) can be found in Table 2. We chose the AAR method as opposed to the Accumulation Area Balance Ratio (AABR) method, even though it is likely the AABR method better-approximates modern glacier ELAs (Pellitero

et al., 2015) for two reasons. First, there are required observations for the AABR method that are difficult to quantify for paleoglaciers, namely the balance ratio, which must be linear for the method to be valid (Osmaston, 2005). Second, since we apply the same exact methodologies for each paleoglacier reconstructed, we assume that any changes to methodologies (e.g., here using AAR vs AABR, using different basal shear stress values, etc.) would result in systematic shifts that would not appreciably impact the magnitude of ELA change observed. As a final observation, a recent study of paleoglacier and ELA

reconstructions Alaska-wide used both AAR and AABR methods for calculating ELAs and found minimal difference in their results (Walcott et al., 2024). All points considered, we elected to use the simpler, but still effective, AAR method.

Finally, we calculate summer temperature anomalies for each glacier position relative to the LIA by applying the dry adiabatic lapse rate of 10°C/km to each ΔELA. The dry adiabatic lapse rate is applied because it serves as an endmember; over timescales

greater than a few hours, it is not possible for an air mass to sustainably exceed the dry adiabatic lapse rate. Thus, our reconstruction likely provides a maximum estimate in the magnitude of temperature change at each glacier position because lower lapse rates would translate to smaller changes in atmospheric temperature.

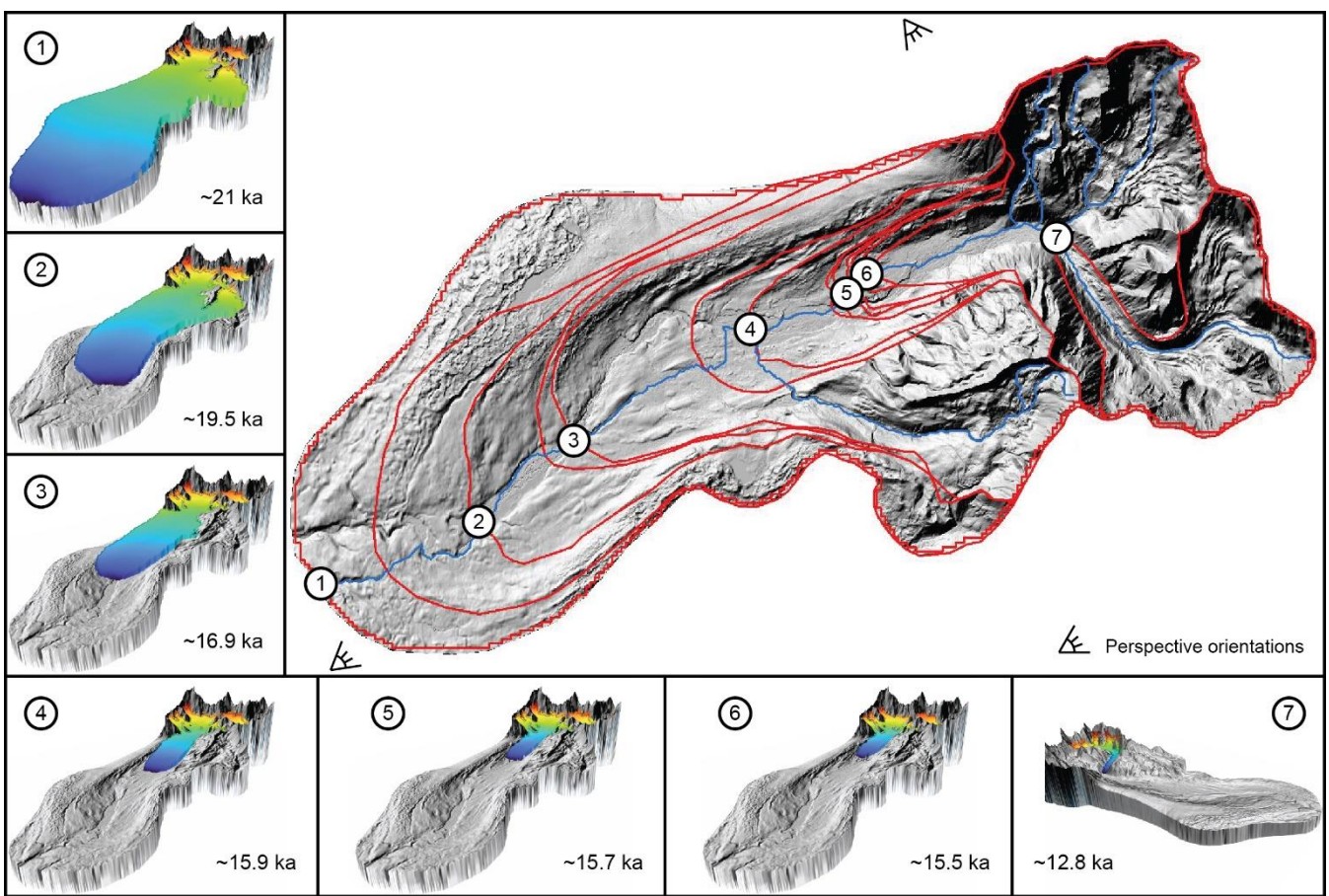

 **Figure 6. Plan view of the North Swift River valley and reconstructed glacier extents in the main panel. Outlines of every glacier surface/extent reconstructed in this analysis (in red). Blue lines are the flowlines reconstructed on the resampled, lower-resolution DEM. In the insets are 3D visualizations of reconstructed glacier surfaces. ELA reconstructions at each glacier position viewed in oblique 3D with 4X vertical exaggeration. Each number in the main panel corresponds to the toe of the glacier position reconstructed in each inset panel. Glacier surfaces 1-6 are viewed from the perspective in the bottom left corner of the main panel and glacier**

**surface 7 is viewed from the perspective near the top right.**

Our ELA reconstructions indicate ~400 m of total ELA rise through the deglacial interval (975 m asl – 1362.5 m asl), which translates to approximately 4°C of warming (Fig. 7, Table 2). While there was notable lateral recession between 21-16 ka, the valley floor here is low-sloping, and we observe minimal ELA rise (~50 m) and modest warming (~0.4°C). Following this

interval, a period of abrupt ELA rise corresponding to ~1.5°C of warming occurred, followed by a pause in deglaciation ~16-15 ka when several moraines were deposited. After this period, a second interval of net ~2°C warming from ~15 to 12.8 ka led to glacier recession and non-deposition of moraines until the final late glacial moraine preserved in our valley was deposited at $12.8 \pm 0.6$ ka with an ELA of 1362.5 m asl, only 16 m below the ELA of the outermost late Holocene moraine in the valley.

**Table 2. Glacier equilibrium line altitude information.**

| Moraine name | Mean ELA | ELA error | Moraine age | Moraine age error |
|:---:|:---:|:---:|:---:|:---:|
| | (m) | (m) | (ka) | (ka) |
| F2-I A | 975 | 16 | 21.3 | 0.8 |
| F2-I B | 1019 | 18 | 20.1 | 1.3 |
| F2-II A | 1028 | 14 | 19.5 | 1.0 |
| F2-II B | 1030 | 14 | 18.7 | 0.8 |
| F2-III A | 1031 | 13 | 16.9 | 1.2 |
| F2-III B | 1043 | 18 | 16.0 | 0.9 |
| F2-III C | 1072 | 20 | 15.9 | 0.9 |
| F2-III D | 1141 | 22 | 16.6 | 0.6 |
| F2-III E | 1152 | 23 | 15.7 | 0.7 |
| F2-IV A | 1172 | 23 | 16.1 | 0.9 |
| F2-IV B | 1175.5 | 23.5 | 15.0 | 0.7 |
| F2-IV C | 1182.5 | 23.5 | 15.5 | 0.7 |
| F2-IV D | 1206 | 26 | 15.7 | 0.7 |
| F2-IV E | 1362.5 | 20.5 | 12.8 | 0.6 |
| LH-1 | 1378.5 | 20.5 | 0.75 | 0.05 |

**Notes: All ELAs are measured using $0.63 \pm 0.04$ Accumulation Area Ratios. All ages reported here are calculated in Tulenko et al. (2022).**

**5 Discussion**

Given that modern average lapse rates in Alaska may be as low as ~4.3°C/km (Verbyla and Kurkowski, 2019), temperature estimates through the last deglaciation could be less than half the magnitude reported here (2°C total warming through deglaciation). However, additional proxy evidence across Alaska spanning deglaciation from pollen assemblages (Viau et al., 2008), fossil chironomids (Kurek et al., 2009), and leaf wax hydrogen isotope data (Daniels et al., 2021) – as well as Gulf of Alaska sea surface temperatures (Praetorius et al., 2020) – suggest ~4°C summer temperature depressions during the LGM, consistent with the glacier-based temperature reconstruction from the Revelation Mountains. Several data assimilation products based on marine sediment records further substantiate the magnitude of cooling observed in our record (Tierney et al., 2020). Moreover, multiple lines of proxy evidence from Alaska indicate conditions were drier than present through much of the last deglaciation until the late YD (Viau et al., 2008; Finkenbinder et al., 2014, 2015; Dorfman et al., 2015), suggesting the time-averaged late Pleistocene environmental lapse rate was likely higher than modern and closer to the endmember dry lapse rate used here. However, there is a notable increase in precipitation observed following the YD in Alaska (e.g., Viau et al., 2008; Kaufman et al., 2010) that could have modulated subsequent glacier advances in the Holocene. Thus, this shift in precipitation may have enhanced the LIA advances at our site to nearly match the youngest late glacial moraine (F2-IV) despite a comparatively warmer Holocene climate. In this scenario, by not accounting for changes in precipitation, we would be underestimating either the magnitude of warming from the late glacial into the Holocene or the magnitude of late glacial cooling relative to the LIA. While we still lean on evidence suggesting that summer temperature is the most important climatic factor for alpine glaciers (Rupper and Roe, 2008), and the total magnitude of warming observed at our site using our end-member lapse rate (10 °C/km) is consistent with other lines of proxy evidence, further examination of the role of precipitation variations through deglaciation and the Holocene on glacier advances at our site and others across Alaska is likely warranted.

Moraine ages from across Alaska reveal an emerging pattern, although in no other valley is the moraine record as complete or well dated as in the Revelation Mountains. The Last Glacial Maximum advance culminated in Alaska ~21-19 ka (Kaufman et al., 2011, Tulenko et al., 2018). Additionally, several moraines have been dated between ~16-15 ka during the later stage of Heinrich Stadial 1 (HS-1). In rare cases, most notably in the Revelation Mountains and in the Ahklun Mountains (e.g., Young et al., 2019), moraines date to ~12.8-12.1 ka during the late glacial and within the earliest and coldest interval of the Younger Dryas (YD). Despite limited chronologic constraints, the magnitude of glacier retreat observed in other valleys in Alaska appears to track with glaciers in the Revelation Mountains (Tulenko et al., 2022). Thus, it is possible the last deglaciation climate reconstruction from the Revelation Mountains may be representative of the broader Alaska region.

Following the last deglaciation, we observe minimal glacier activity at our site through much of the Holocene until the LIA, when glacier advances nearly matched the extent of the youngest late glacial advance. While this is consistent with glacier behavior at several sites in Alaska, we note the contrasting pattern of Holocene glacier activity observed in valleys in the

Brooks Range. Observations suggest that some neoglacial advances predating the LIA were slightly more extensive in the region (e.g., Ellis and Calkin, 1984; Badding et al., 2013; Pendleton et al., 2017). Moreover, in the Ahklun Mountains, Young et al. (2019) observe two minor glacier re-advances in the Waskey valley at 12.5 ± 0.1 ka and 12.1 ± 0.4 ka. While there appears to be no preserved evidence of glacier advances between the 12.8 ± 0.6 ka moraine and the LIA moraines in the

Revelation Mountains, it is possible that minor advances occurred and were overridden by subsequent advances during the LIA. Regardless, late glacial-to-Holocene moraines deposited in Alaska narrowly exceed LIA advances indicating – without considering the role of precipitation – that late glacial and early-to-mid Holocene cooling events were minor in Alaska. While this observation may appear at odds with the magnitude of North Atlantic climate oscillations observed in Greenland Ice Core records (Fig. 7), mounting evidence suggests these oscillations may be more strongly expressed during winter compared to

summer (e.g., Bromley et al., 2018). Because there is likely a heavy summer seasonal bias to alpine glacier-based temperature reconstructions, we suggest that predominantly winter-time North Atlantic oscillations would result in a somewhat dampened response from alpine glaciers, which is consistent with observed minor late glacial cooling at our site. Indeed, there is evidence from the southwestern Greenland Ice Sheet margin suggesting that the margin was retreating during the YD in response to summertime and local feedbacks, seemingly in contrast to the notable cooling observed in the Greenland Ice Core records

(Funder et al., 2021; Carlson et al., 2021). This observation might further support the notion of YD cooling being a predominantly winter phenomenon.

In the context of detailed glacier reconstructions observed elsewhere around the world, we find both similarities and differences between our record and others (Fig. 7). Broadly, Southern Hemisphere glacier records indicate a coupling with global $CO_2$

reflected in Antarctic ice core records, while glaciers in Alaska follow patterns of alpine glacier recession in the Alps and climate fluctuations recorded in Greenland Ice Cores (Fig. 7). Even if rising global $CO_2$ forced major recession following deposition of the 17-ka moraine (F2-IIIA) and any evidence was overridden by subsequent moraine deposition events at our site, that scenario would still require a large magnitude re-advance during HS-1 in Alaska based on our record. While evidence for YD advances in Alaska extend beyond our site (Young et al., 2019), we acknowledge that within the resolution of [10]Be

dating, the late glacial moraine at our site may have been deposited at the time of the Antarctic Cold Reversal 'breakpoint' like records in the Southern Hemisphere and a site in Norway (Putnam et al., 2023). Despite uncertainty in the age assignment of one moraine at our site, we still observe a dampened climate response in Alaska at intervals generally tracking North Atlantic climate oscillations – cooling and moraine deposition at the culmination of HS-1, retreat through the Bølling warm period and slight cooling during the early YD (Fig. 7) – with substantial deviation from records in the Southern Hemisphere, which all

reinforces the notion that the North Atlantic signature of climate evolution through deglaciation extended out to Alaska.

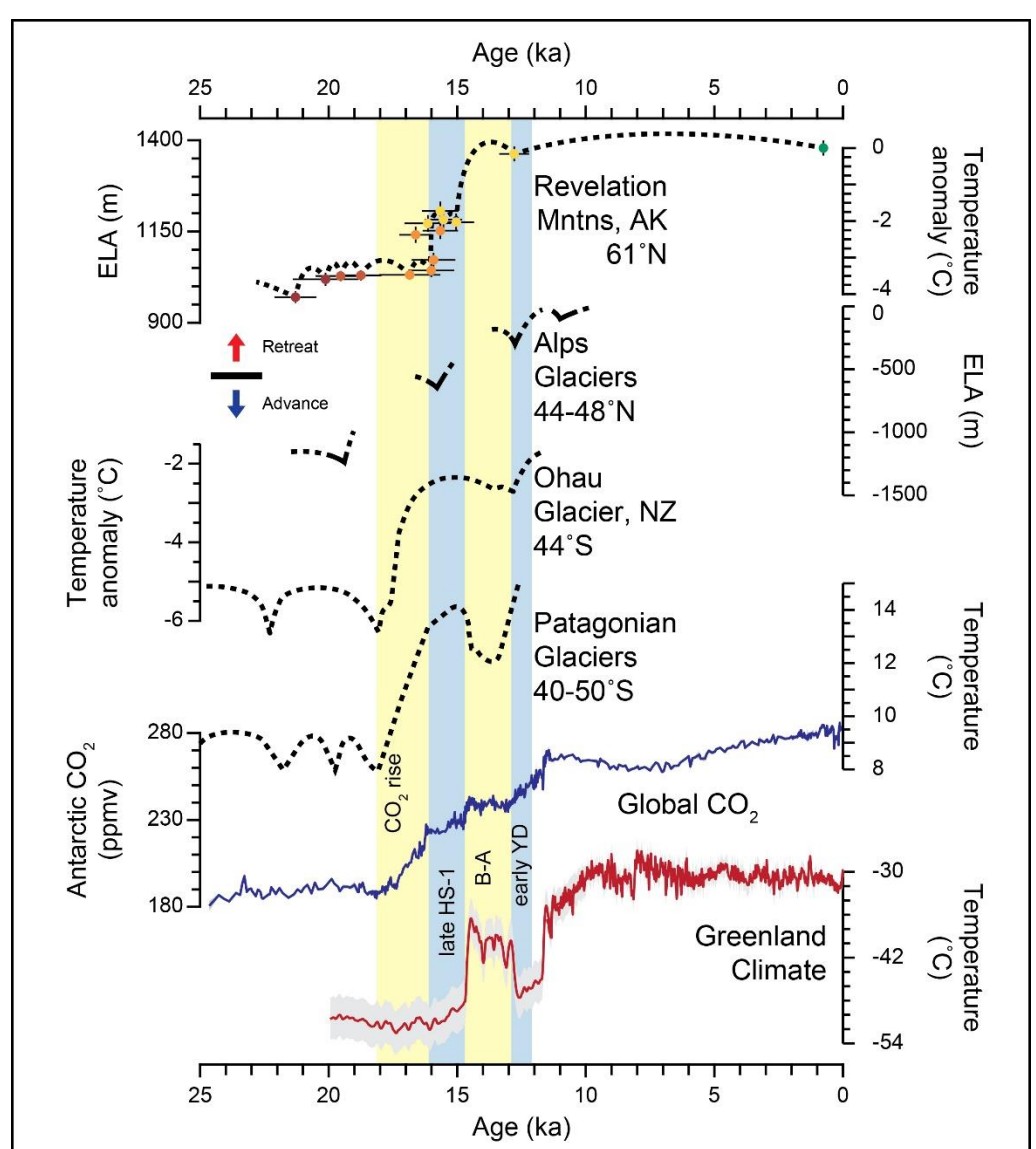

**Figure 7. Comparison of the Revelation Mountains retreat chronology with Southern Hemisphere and Alps glacial retreat, and global/regional climate forcing mechanisms. Top to bottom: the Revelation Mountains ELA-based temperature curve (this study), composite ELA reconstructions from the Alps (Ivy-Ochs, 2015), temperature curve from Ohau Glacier, NZ (Putnam et al., 2013), composite temperature curve from Patagonian Glaciers (Denton et al., 1999; Strelin et al., 2011), composite atmospheric $CO_2$ concentrations from Antarctic Ice Cores (Bereiter et al., 2015), and Greenland-wide average mean annual temperature from ice cores (20-11 ka; Buizert et al., 2014, 11-0 ka; Kobashi et al., 2017).**

## 6 Conclusions

New [10]Be ages from late Holocene moraines in the Revelation Mountains reveal similar culminations with other, well-dated records in southern Alaska at ~1200-1300 CE and ~1700 CE, suggesting similar climatic controls across southern and western

Alaska. The ELA-based temperature record from the Revelation Mountains, pinned by the late Holocene moraine constraints, indicates an LGM temperature depression of ~4°C relative to the LIA, consistent with other limited proxy data from across the state. Most warming was delayed following global $CO_2$ rise until ~16 ka when abrupt, ~1.5°C of warming forced significant retreat. Brief intervals of cooling and moraine deposition late during HS-1 and in the early YD interrupted the overall pattern of warming and glacier retreat, similar to the observed glacial records in the Alps. Paired with ~2°C of warming and glacier recession during the Bølling, these observations highlight the potential influence of North Atlantic forcing on climate in Alaska.

**Data availability**

All observations necessary to re-calculate $^{10}$Be concentrations and exposure ages are reported in the main text, as well as in the ICE-D online database (www.ice-d.org).

**Author Contribution**

JPT was responsible for conceptualization, formal analyses, visualizations and writing of the original draft for this manuscript. JPB, NEY, and JMS provided resources, funding acquisition, conceptualization, and review and editing of the original draft.

**Competing interests**

The authors declare that they have no conflict of interest.

**Acknowledgements**

We would like to acknowledge that samples in this study were collected on ancestral lands of the Dënéndeh and Dena'ina Ełnena and that the University at Buffalo exists on the land of the Seneca. These peoples are the traditional caretakers of these lands. We give thanks for the opportunity to exist and work on lands that are rightfully theirs. We thank R. Schwartz for lab processing and LLNL for $^{10}$Be measurements. Young and Schaefer acknowledge support from the LDEO Climate Center. This manuscript was supported by NSF grant 2135466.

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
