# Peer review of "Abrupt warming and alpine glacial retreat through the last deglaciation in Alaska interrupted by modest Northern Hemisphere cooling"

_Climate of the Past, 2023_

## Author Response (AR1)

**Response to Reviewer 1**

We thank RC 1 for their thoughtful and constructive review of our manuscript. We have gone through and replied to each individual comment. Below, please find the original comments bolded and italicized and our reply in normal font.

***There a few small areas where the manuscript could in my opinion be improved. First, I would like there to be a table that includes the data to calculate nuclide concentrations. Secondly, I would like to see a visualization to help assess moraine age robustness and outliers. I suggest adding a PDF plot that shows all the samples from the late Holocene moraines at once. Third, I also make some notes about the role of precipitation in the Holocene versus the deglacial period in Alaska that comes up in the discussion. Overall, I think the authors do a good job in the discussion of accounting for the most obvious line of contention: the adiabatic lapse rate and how it may have changed through time.***

We appreciate the opportunity to update the data table provided since, as the reviewer suggested below, there is room for improvement. A summed pdf of all exposure ages in the study can be added to section 3. In regard to differences in precipitation between the deglacial and Holocene, the points highlighted below by the reviewer are important and will be addressed in more detail in the revised manuscript.

***Line-By-Line Comments***

***Line 58. 'South' should be lowercase.***

Fixed

***Line 64 and 66. I'm not totally clear on the use of the word 'surveyed' in this paragraph, really between Line 64 and Line 66. "After a thorough moraine dating campaign of deglacial moraines in the valley, we further surveyed…" It seems like the first usage is about selecting a moraine series to date, whereas the second usage on Line 66 is about a more quantitative surveying, namely mapping and dating. Just a note.***

We agree the word choice is somewhat confusing here, so we updated the sentence starting at line 65 to say:

"After a thorough moraine dating campaign of deglacial moraines in the valley, we additionally mapped and dated moraines in the valley deposited near the terminus of the extant glacier that were likely late Holocene in age (Fig. 2)."

***Line 70. Blending Methods and Results is fine by me in Sections 3 and 4. That said, could you mention in the subheadings for Sections 3 and 4 something about the Methods? E.g. 'Cosmogenic exposure dating and late Holocene glacier fluctuations in Alaska' or 'ELA Modeling and climate change in the Revelation***

*Mountains through deglaciation' (and variations more creative than that). Would help for organizational clarity.*

We updated the header for section 3 to read:

"Cosmogenic ${}^{10}$Be exposure dating of late Holocene glacier fluctuations in the Revelation Mountains"

We also updated the header for section 4 to read:

"Glacier surfaces, ELAs and climate reconstructions in the Revelation Mountains through deglaciation"

*Line 82 (Figure 2). As someone who works in the Holocene, I find the convention of using calendar dates in Figure 2 confusing. I would prefer to see the ages in annums or ka. The ages that go negative are quite difficult to understand, and commonly the convention is to present as exposure duration. I don't mind using calendar dates in the text, however, especially in the Discussion when other climate records are brought in (in fact it's quite helpful there). Consistency between the 'Younger Dryas' moraine (F2-IV E, reported in exposure duration) and the Holocene moraines (reported in calendar dates) is another reason to stick to exposure duration. It is also how results are presented in Table 2. As a suggestion, I've seen other deglacial to Holocene papers where 'ka' is used for ages > 1,000 years and 'a' for ages < 1,000 years and liked that convention. Up to the authors and editor, however.*

Ages reported in the figure are now reported in annum instead of common era.

*Line 93 (General comment on data replicability). I appreciate that all the samples are already on ICE-D and age recalculation is easy. However, there does not seem to be any of the raw data necessary if I want to recalculate the concentration of 10Be atoms in the sample, which is necessary for verification. In particular, the reported uncertainties seem low to me; it may be completely fine, but as is I cannot recalculate nuclide concentrations and independently verify their work, which is one of my objectives when I review a paper. Could the authors add a table (to a supplement is fine) with the requisite information? Data such as 10/9Be ratio from the AMS, grams of quartz used (7 g versus 40 g will impact measurement uncertainty, which I currently cannot evaluate), carrier concentration, blank 10/9Be ratios, etc. It ensures the longevity of the data and is necessary for accordance with standards for geochemical labs, data replicability, etc.*

We thank the reviewer for highlighting an opportunity for us to be more transparent with our data. This is important and we agree that the table needs to be updated. We chose to incorporate sufficient sample information directly into this table to not only recalculate

exposure ages, but also recalculate Be-10 concentrations. This includes carrier information, quartz dissolved and AMS ratio information.

***Line 94. Would it be possible to put Table 1 here? Near Line 108 would also work for me. It feels out of order as is. It's useful for interpreting the paragraph from Lines 108–116, so at least right after that paragraph I'd find helpful.***

Table 1 has been moved up to this location.

***Line 102. This was said already, or at least nearly so, on Lines 78–81. Consider combining these and streamlining.***

We chose to omit the redundant portion of the clause and move the "wind-swept" phrasing to the lines 78-81 in the previous section.

***Line 106. Circling back to exposure ages vs. calendar date, I've thought about this a good bit and think it is important the data are reported as exposure ages, simply because a year in time is not an exposure age. Here is my logic: The convention is to report ages in years of exposure, or exposure age. That differs from specifying a point in time in which samples were exposed, or a calendar year. Since the measurements are referred to as 'exposure ages,' they should be reported as exposure duration, in years or annums. At the very least, this is what 'exposure age' implies in the literature. The authors are absolutely free to provide calendar dates in the parenthetical (like on Line 108) and Table 1, or even completely switch to that format once they provide the exposure ages. Again, I think it is even helpful to do so for comparison with the dendrochronology data, and a good idea by the authors. As written, I am not sure it is correct, but either way I think it would improve the manuscript to start with exposure duration before switching to calendar year of exposure.***

Here and throughout, we elected to report ages in annum in the text while also keeping the ages reported in common era in parentheses.

***One other note on this subject is that the date of sampling is not presented in the paper. I see from ICE-D that it was in 2019, but with the calendar year, is that value being subtracted from 2019? (I think so, but then there's radiocarbon's conventions it being reported relative to 1950, etc etc. The clarity will help, that is all.) Would be worth saying 'samples were collected in 2019' around Line 71.***

Agreed it would be helpful to specify in the paper when samples were collected so readers know 'exposure ages' are relative to 2019 and not, for example 1950 like with calibrated radiocarbon conventions. We updated the sentence starting at line 71:

"To complete the time series of glacier retreat in the North Swift River valley, we collected surface samples in summer 2019 for $^{10}$Be dating from 15 granitic boulders…"

We also entered a sentence at line 101 to clarify 2019 as the reference starting point for our CE conversion:

"Ages from this study converted to common era (CE; below) use 2019 (date of collection) as the reference starting date."

**Lines 110----116. I find it difficult to evaluate the robustness of the moraine ages as presented. I request that the authors add a PDF plot that shows all the data at once. In Figure 5, the older outliers are not on the plot. I think this is an important plot to add, since n = 5 on each moraine and on one moraine (LH-2), 3 of 5 ages are discarded as outliers. A PDF plot could go here or after Lines 150—155.**

We thank the reviewer for the suggestion. Instead of creating a whole new figure, we elected to add a B panel to figure 2 that has a pdf plot for each individual age and summed pdf for the clusters of ages.

**Line 119 (Figure 3). Same thing, I recommend switching to years of exposure.**

Updated.

**Lines 149—150. Citation needed.**

We cite Heyman et al. (2011) here.

Heyman, J., Stroeven, A. P., Harbor, J. M., and Caffee, M. W.: Too young or too old: evaluating cosmogenic exposure dating based on an analysis of compiled boulder exposure ages, Earth and Planetary Science Letters, 302, 71–80, 2011.

**Line 189 (Figure 4). Might be nice to add the moraine ages in the numbered side-panels. Helps for interpretation. Just some small text with age and uncertainty in the lower-right-hand corner of each panel, for example. This is subjective, though; feel free to ignore.**

We chose to add Approximate ages to the corner of each panel.

**Lines 221–226. I'm not sure about this. This gets into my general comment about the role of precipitation. While I do think the authors do a good job addressing the elephant in the room (adiabatic lapse rate changing between Deglacial and Holocene), I had to look into the references to fully understand the authors' point here. Could you add a sentence or two clarifying what specifically you draw from these references? I discuss this further in the next line comment for Line 246…**

**Lines 246–247. This a surprising result, really. Commonly the YD is considered much colder than the LIA; this is clear from the Greenland ice core record the authors plot in Figure 7. To me, it is completely possible that a wetter Holocene may explain why the YD moraine and the late Holocene moraine are only 0.1° C**

*apart in temperature space (assuming no precip change). A colder, dryer deglacial leading to a warmer but wetter Holocene could potentially cause advances that approach the YD moraines. I think it matters which way the authors are interpreting the precipitation change during the YD and into the Holocene, and therefore my only request is a bit more clarity to this end. I acknowledge that the authors make clear their preference for interpreting ELA change as overwhelmingly summer temperature, but this is not agreed upon in the literature. Sea level (source moves closer), ocean circulation, and the jet stream changed dramatically from the deglacial to the Holocene; it is not inconceivable to me that precipitation could be significantly different. The authors support their claims well and I have no problem with their interpretation; it is what the Discussion section is for.*

*The authors prefer ~4° C LGM to LIA change because it matches other local records, but as they correctly point out, using a more standard lapse rate lowers this value. To me, a significantly different precipitation regime in the Holocene could explain this mismatch.*

The three comments above are important points that we agree should be addressed in our discussion section. While ultimately the total magnitude of temperature change through the deglacial period using our end-member lapse rate is consistent with other proxy evidence, it is fair to argue that a wetter Holocene could have modulated the LIA advance and impacted our point of reference for the magnitude of temperature rise and retreat through deglaciation compared to the Holocene. We updated the section beginning at line 221 with the following:

"However, there is a notable increase in precipitation observed following the YD in Alaska (e.g., Viau et al., 2008; Kaufman et al., 2010) that could have modulated subsequent glacier advances in the Holocene. Thus, this shift in precipitation may have enhanced the LIA advances at our site to nearly match the youngest late glacial moraine (F2-IV) despite a comparatively warmer Holocene climate. In this scenario, by not accounting for changes in precipitation, we would be underestimating either the magnitude of warming from the late glacial into the Holocene or the magnitude of late glacial cooling relative to the LIA. While we still lean on evidence suggesting that summer temperature is the most important climatic factor for alpine glaciers (Rupper and Roe, date), and the total magnitude of warming observed at our site using our end-member lapse rate (10 °C/km) is consistent with other lines of proxy evidence, further examination of the role of Holocene precipitation on glacier advances at our site and others across Alaska is likely warranted."

In response to the somewhat surprising conclusion we draw about YD cooling being very minor at our site, we agree that the section beginning at line 247 needs the following clarification/revision:

"Regardless, late glacial-to-Holocene moraines deposited in Alaska narrowly exceed LIA advances indicating – without considering the role of precipitation – that late glacial and

early-to-mid Holocene cooling events were minor in Alaska. While this observation may appear at odds with the magnitude of North Atlantic climate oscillations observed in Greenland Ice Core records (Fig. 7), mounting evidence suggests these oscillations may be more strongly expressed during winter compared to summer (e.g., Bromley et al., 2018). Because there is likely a heavy summer seasonal bias to alpine glacier-based temperature reconstructions, we suggest that predominantly winter-time North Atlantic oscillations would result in a somewhat dampened response from alpine glaciers, which is consistent with observed minor late glacial cooling at our site."

*Line 250. Same thing, can you explain this? 1 more sentence maybe. If the YD is much colder than the LIA in the North Atlantic (Figure 7), why does your record—which suggests the LIA is nearly as cold as the YD—suggests North Atlantic climate forcings extend to Alaska? Later (Line 261) you say the response is dampened. What's written is not necessarily in conflict, just seeking some clarity.*

Hopefully the revision/clarification written in response to the previous comment will clarify our point here.

*Line 269 (Figure 7). I'm not sure about the dashed lines temperature reconstructions in Figure 7. Or said another way, the Revelations reconstruction has the actual moraine ages plotted and the dashed lines superimposed on it, which is reasonable to me. The Alps data too seems to have minimal inference between points and clearly lays out the data (moraines, which ELAs are derived from) as a solid line vs interpretation as a dashed line. The Ohau Glacier and Patagonian glaciers are only the dashed lines. Can you make this consistent between the plots? Something to make the Ohau and Patagonia more like the Revelations data or the Alps data.*

Thank you for the suggestion. Based on our literature review, we find it's likely there is comparable certainty in moraine ages in the SH sites compared to the sites in the Alps, so we have elected to update the figure with solid lines where moraines are plotted for the SH plots.

*I also think the paper would be strengthened by including some sort of precipitation data in Figure 7, perhaps the data referenced in Viau et al., 2008 or Kaufman et al., 2010. This is not a requirement, just a suggestion.*

We thank the reviewer for this suggestion, but since we have not quantifiably assessed the influence of precipitation on our record, we feel this is outside the scope of this manuscript and could be investigated in later studies.

**Response to Reviewer 2**

We thank RC 2 for their thoughtful and constructive review of our manuscript. We have gone through and replied to each individual comment. Below, please find the original comments bolded and italicized and our reply in normal font.

***In the glacier/ELA modelling several methodological decisions are made without being clearly explained and without the impacts of these decisions (on study results) being tested. For example, ELA is calculated using the AAR method. However, several studies highlight the merits of using the AABR method instead (i.e. it better approximates measured ELAs for modern glaciers). I suggest calculating ELA using the AABR (alongside AAR if required) at least to demonstrate the impact this has on results.***

We agree that the AABR method does a better job of approximating ELAs for modern glaciers, but the method requires a larger amount of observations that are particularly challenging to make for paleoglaciers. We add the following statement at line 183 to justify the use of the simpler, AAR method:

"We chose the AAR method as opposed to the Accumulation Area Balance Ratio (AABR) method, even though it's likely the AABR method better-approximates modern glacier ELAs (Pellitero et al., 2015) for two reasons. First, there are required observations for the AABR method that are difficult to quantify for paleoglaciers, namely the balance ratio, which must be linear for the method to be valid (Osmaston, 2005). Second, since we apply the same exact methodologies for each paleoglacier reconstructed, we assume that any changes to methodologies (e.g., here using AAR vs AABR, using different basal shear stress values, etc.) would result in systematic shifts that would not appreciably impact the magnitude of ELA change observed. As a final observation, a recent study of paleoglacier and ELA reconstructions Alaska-wide used both AAR and AABR methods for calculating ELAs and found minimal difference in their results (Walcott et al., 2024). All points considered, we elected to use the simpler, but still effective, AAR method."

Walcott, C. K., Briner, J. P., Tulenko, J. P., and Evans, S. M.: Equilibrium line altitudes of alpine glaciers in Alaska suggest Last Glacial Maximum summer temperature was 2–5\,\degreeC lower than during the pre-industrial, Climate of the Past, 20, 91–106, https://doi.org/10.5194/cp-20-91-2024, 2024.

***On page 10, you mention setting the basal shear stress to 1 hPa. I presume this value is 100 kPa? In addition, there is no mention of the sensitivity of results to different shear stress values.***

That is correct, the value we chose is 100 kPa. We have changed the units in the manuscript to say 100 kPa for clarity. As highlighted above, we suggest that any changes to this value applied to every glacier surface would produce systematic changes that would presumably not impact the magnitude of ELA change through deglaciation.

***The paper's focus is temperature reconstruction. Past precipitation is mentioned
but only really to justify the choice of lapse rate. In practice, if past precipitation
differed from present and/or changed throughout deglaciation then the
temperature reconstructions presented here would be quite different (since
precipitation impacts glacier dimensions). I feel there is scope for further
consideration of this impact.***

We thank the reviewer for highlighting this observation. It is something reviewer 1 also
highlighted, and we hope that the edits to the manuscript made in response to their
comments cover the concerns presented here as well. Our response to reviewer 1 is
reiterated here:

While ultimately the total magnitude of temperature change through the deglacial period
using our end-member lapse rate is consistent with other proxy evidence, it is fair to
argue that a wetter Holocene could have modulated the LIA advance and impacted our
point of reference for the magnitude of temperature rise and retreat through deglaciation
compared to the Holocene. We updated the section beginning at line 221 with the
following:

"However, there is a notable increase in precipitation observed following the YD in
Alaska (e.g., Viau et al., 2008; Kaufman et al., 2010) that could have modulated
subsequent glacier advances in the Holocene. Thus, this shift in precipitation may have
enhanced the LIA advances at our site to nearly match the youngest late glacial
moraine (F2-IV) despite a comparatively warmer Holocene climate. In this scenario, by
not accounting for changes in precipitation, we would be underestimating either the
magnitude of warming from the late glacial into the Holocene or the magnitude of late
glacial cooling relative to the LIA. While we still lean on evidence suggesting that
summer temperature is the most important climatic factor for alpine glaciers (Rupper
and Roe, date), and the total magnitude of warming observed at our site using our end-
member lapse rate (10 °C/km) is consistent with other lines of proxy evidence, further
examination of the role of precipitation variations through deglaciation and the Holocene
on glacier advances at our site and others across Alaska is likely warranted."

***Linked to the above, presumed former aridity is used to justify a large (ish) lapse
rate, but there is little mention of the impact this continentality might have on
glacier dimensions. Where the amplitude of annual temperatures varies through
time (i.e. where continentality changes), glacier dimensions can fluctuate without
any corresponding changes in summer temperatures (see Golledge et al., 2010). I
think this is something to discuss/consider.***

We thank the reviewer for bringing this observation to our attention. While we still feel
confident in our reconstruction based on the coherence of our record with other proxy
evidence, this is still something worth noting, which we state in the concluding sentence
above.

***Golledge, N., Hubbard, A. and Bradwell, T., 2010. Influence of seasonality on glacier mass balance, and implications for palaeoclimate reconstructions. Climate Dynamics, 35, pp.757-770.***

**Response to the editor**

We thank Dr. Reyes for handling our manuscript and for providing additional feedback on one of our responses. Please find the original feedback from Dr. Reyes below bolded and italicized and our response in normal font.

***Regarding the relatively dampened YD response of alpine glaciers (at least, relative to the proxy-inferred and possibly winter-biased MAT records), there's potential for additional context on this from west Greenland where a few recent studies (Carlson et al QSR 2021; Funder et al ClimPast 2021) document ice-margin retreat during the YD. Full disclosure that I am a co-author on one of those papers. You and your co-authors may decide this is a little outside the context of the manuscript or not a great comparison due to ice-sheet vs alpine-glacier margins; I only bring it up because one of your responses to Rev 1 comments raises the issue. No obligation whatsoever to incorporate this.***

Thank you for bringing this up, this is another interesting observation that provides further evidence that YD cooling observed in Greenland Ice Core records might be a predominantly winter time phenomenon. We feel this is worth noting in our manuscript, so we added the following sentences after the revised passage we added in response to reviewer 1 comments:

"Indeed, there is evidence from the southwestern Greenland Ice Sheet margin suggesting that the margin was retreating during the YD in response to summertime and local feedbacks, seemingly in contrast to the notable cooling observed in the Greenland Ice Core records (Funder et al., 2021; Carlson et al., 2021). This observation might further support the notion of YD cooling being a predominantly winter phenomenon."